# Repeatability of flatfish reflex impairment assessments based on video recordings

**Sven Sebastian Uhlmann**[1]*, **Noëlle Yochum**[2], **Bart Ampe**[1]

**1** Flanders Research Institute for Agriculture, Fisheries and Food, Animal Sciences Unit, Ostend, Belgium,
**2** NOAA, Alaska Fisheries Science Center, Seattle, United States of America

* sven.uhlmann@gmx.net

## Abstract

Using measures of reflex impairment and injury to quantify an aquatic organism's vitality have gained popularity as survival predictors of discarded non-target fisheries catch. To evaluate the robustness of this method with respect to 'rater' subjectivity, we tested inter- and intra-rater repeatability and the role of 'expectation bias'. From video clips, multiple raters determined impairment levels of four reflexes of beam-trawled common sole (*Solea solea*) intended for discard. Raters had a range of technical experience, including veterinary students, practicing veterinarians, and fisheries scientists. Expectation bias was evaluated by first assessing a rater's assumption about the effect of air exposure on vitality, then comparing their reflex ratings of the same fish, once when the true air exposure duration was indicated and once when the time was exaggerated (by either 15 or 30 min). Inter-rater repeatability was assessed by having multiple raters evaluate those clips with true air exposure information; and intra- and inter-rater repeatability was determined by having individual raters evaluate a series of duplicated clips, all with true air exposure. Results indicate that inter- and intra-rater repeatability were high (intra-class correlation coefficients of 74% for both), and were not significantly affected by background type nor expectation bias related to assumed impact from prolonged air exposure. This suggests that reflex impairment as a metric for predicting fish survival is robust to involving multiple raters with diverse backgrounds. Bias is potentially more likely to be introduced through subjective reflexes than raters, given that consistency in scoring differed for some reflexes based on rater experience type. This study highlights the need to provide ample training for raters, and that no prior experience is needed to become a reliable rater. Moreover, before implementing reflexes in a vitality study, it is important to evaluate whether the determination of presence/absence is subjective.

## Introduction

To address concerns over discard practices and animal welfare in commercial fisheries, methods are needed to reliably profile fish condition onboard vessels to describe fishing impacts on both individuals and populations [1–4]. To allow assessments in adverse and remote conditions, responsiveness of a fish to induced stimuli expressed as a binary presence-absence score

**Data Availability Statement:** The relevant source data are available from the Marine Data Archive (http://mda.vliz.be); https://doi.org/10.14284/399.

**Funding:** The work was supported by the European Maritime and Fisheries Fund (Grant Numbers: VIS/

14/B/01/DIV, 17/UP1/01/Div, and 18/UP1/30/Div).
The funder had no role in study design, data
collection and analysis, decision to publish, or
preparation of the manuscript.

**Competing interests:** The authors have declared
that no competing interests exist.

may be measured as part of the Reflex Action Mortality Predictor Method (RAMP) [5]. This method assumes that physical stress from mechanical interaction with fishing gear during a capture event may trigger internal physiological responses and alters metabolism (i.e., from anaerobic exercise and hypoxia) which then may precipitate in impaired responsiveness, because the pathway of nerve impulses from the receptors to the muscles through the brainstem and/or the spinal cord might be affected [6–7].

To collect such fish condition or vitality information, some methods involve observers (or so-called 'raters') scoring or rating the extent of external injury and/or responses to stimuli (e.g., reflex impairment), typically using a binary or ordinal scale [7–8]. Depending on the scope of the study, these assessments may be done by different and possibly independent raters with different technical experience and levels of training [4]. When multiple raters are involved in the assessment, rater subjectivity has the potential to affect the accuracy and precision of the vitality assessment [9]. A rater's 'score' may be influenced by (i) knowing the treatment an animal has received (e.g., expectation bias from a non-blinded experimental design; [10–15]; (ii) their level of experience (scientific background or familiarity with vitality scoring), which may lead to a subjective interpretation of otherwise objective scoring criteria [13, 16–17]; and/or (iii) an assessment criterion or metric that is difficult to discern or that lends itself to subjectivity [14, 17–18]. When using vitality scoring, it is important to evaluate whether the assessment criteria are unbiased. This has been highlighted in studies on fish discard mortality [19] as well as in the medical field [17].

Although (semi-) quantitative condition indicators such as reflex impairment ratings have been collected for fishes (e.g., [4–5, 20–21]) and invertebrates (e.g., [22–25]) around the world (largely as a predictor for fisheries discard survival), their sensitivity towards rater biases has not been thoroughly tested. While [9] showed that three raters were able to similarly score reflexes of European plaice (*Pleuronectes platessa*), the role of expectation bias, together with experience intra-rater reliability (also termed 'repeatability'; [11, 26–27] has not been tested for any fish species before.

To address uncertainty in inter- and intra-rater repeatability, we evaluated reflex scores of common sole (*Solea solea*) in the Belgian flatfish beam-trawl fishery. We conducted workshops whereby raters with different backgrounds and levels of scientific experience (i.e., third year veterinary medicine students, practicing veterinarians/food safety inspectors, and fisheries scientists with varying amounts of experience in fish reflex testing) were asked to score fish for reflex impairment (four different reflexes) through video clips using a tagged analogue continuous scale (tVAS; [9]).

Through this study, we aimed to (i) quantify the effect of expectation bias in reflex impairment ratings by testing whether falsified air exposure information misled raters (i.e., evaluating whether informing raters that the animal had prolonged air exposure would bias them toward either higher or lower scores) compared to the duplicated clip (false vs. true air exposure information); (ii) evaluate intra-rater repeatability and the influence of experience, by observing if the same rater was able to reproduce a given score from a duplicated clip (only true air exposure information); and (iii) evaluate inter-rater repeatability among rates for the same clip (including clips with only true air exposure information).

## Materials and methods

### Ethics statement

The handling of animals, including those that were filmed in this research, was approved by the animal ethics commission of the Flanders Research Institute for Agriculture, Fisheries and Food (ILVO, Ref. no. 2016/264). Experiments were performed on-board a commercial Belgian

beam-trawler, the R/V *Simon Stevin* and at a research laboratory in Ostend, Belgium. All research-related handling was designed to minimize any stress cumulative to being captured by beam trawls and sorted on deck. For example, any air exposure during the reflex tests was kept to a minimum and was well within exposure times during conventional, commercial sorting practices. If fish were held captive, housing mimicked natural conditions. The filming did not require any extra handling procedures. Animal ethics approval was granted by the Flanders Research Institute's for Agriculture, Fisheries and Food (ILVO) Animal Care and Ethics Committee (EC2016/264).

## Equipment and treatments

To evaluate inter- and intra-rater repeatability, we conducted a series of seven workshops where separately either third-year, veterinary medicine students, practicing veterinarians/food safety inspectors, or fisheries scientists scored four reflexes of common sole from short (<30 s) video sequences (or 'clips'; Fig 1).

The first scoring session, conducted in April 2015, was attended by third-year, veterinary medicine students from the University of Ghent (N = 120 female and N = 35 male students; N = 2 male non-student experts). The second session, in May 2015, occurred during a lunchtime seminar with fisheries research scientists, with diverse expertise (N = 7 female and N = 11 male). The third session, in December 2015, was during an international workshop of fisheries research scientists with specialist expertise in discard survival studies (N = 5 female, N = 8 male). The fourth session, in January 2016, included seagoing fisheries observers and fisheries technicians (N = 6 female, N = 13 male). The final sessions (5–7) were shown: (5) in April 2016 to third-year veterinary medicine students from the University of Ghent (N = 140 female, N = 39 male students; N = 2 genderless; N = 1 male expert); (6) in December 2016 to practicing veterinarians/food inspectors (N = 12 female, N = 20 male; N = 1 male expert); and (7) in December 2017 to fisheries research scientists with diverse expertise (N = 6 female; N = 7 male; N = 1 genderless).

The reflexes that were selected (body flex, righting, head, and tail grab; Table 1) were those used to assess common sole [4] and that were clearly visible in a video clip. Each workshop began with a 15-min lecture with visual aids detailing the utility of the reflex scoring method as an animal welfare indicator and predictor for discard survival (Supporting Video 1, https://doi.org/10.14284/399). Participants were also informed about relevant factors in the catch-and-discarding process that potentially stress fish and result in weaker reflex responses, namely prolonged periods of air exposure on deck, among others. Following the lecture, participants were trained on example video clips showing, for each of the four reflexes, an 'absent', 'weak', 'moderate', or 'strong' reflex response (Table 1; Supporting Video 1, https://doi.org/10.14284/399). The key criterion associated with each response category was read out loud and given to each participant in the form of a pictogram handout (Fig 2A). These training clips were unique and not used again within the video clips that were scored by the participants.

Video clips were used to test whether the same rater ('intra-rater repeatability'), or different raters ('inter-rater repeatability') were able to repeat the same score of the same fish, and whether a rater's score could have been influenced by knowing how much a given fish was exposed to air prior to its reflex test (intra-rater repeatability with testing an expectation bias effect). A total of 36 video clips of the four reflex responses of common sole were picked out of a reference library of video clips, representing a range of impairment states filmed inside a laboratory (N = 5 fish), or on-board a commercial beam trawler (N = 5 fish; Table 2). Overall, clips included reflexes across the categorized spectrum of responses (ranging from absent to strong). Three experienced expert raters who were involved in the development of the reflex

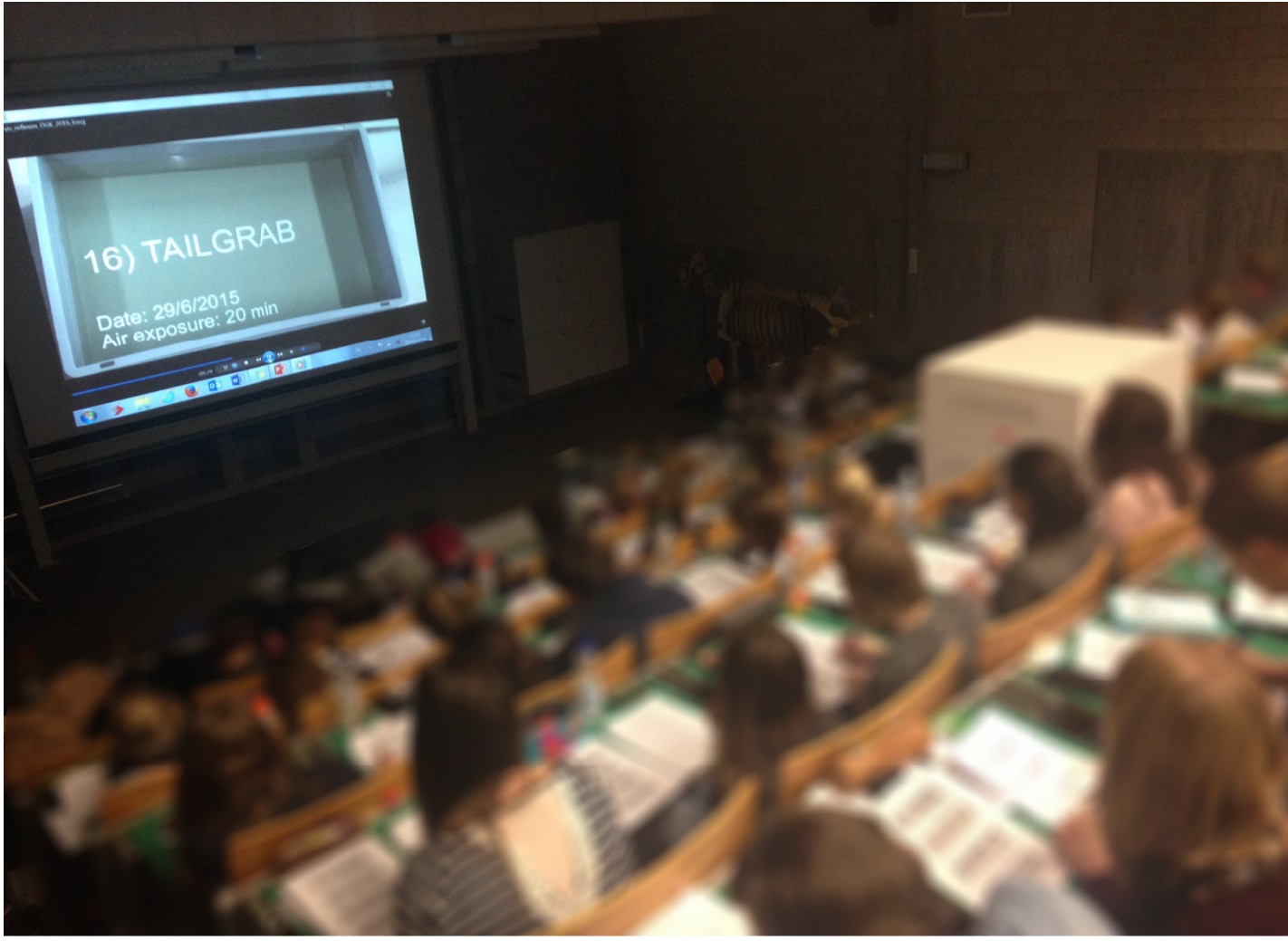

**Fig 1. Lecture theatre view to illustrate video projection.** Third-year veterinary medicine students from the University of Ghent independently scored a tail grab reflex response of a common sole (*Solea solea*) from a video clip projected onto a lecture theatre screen during a workshop session in April 2016. Note: The picture was blurred in parts to guarantee that any person in the audience cannot be identified to comply with the PLOS open-access (CC-BY) license.

scoring methodology scored the 12 unmodified, original clips that were used to create the scoring video used during sessions 2–4 (Fig 3). It showed that the selected clips did not bias reflexes towards either weak or strong responses.

To address intra-rater repeatability and bias related to expectation of an effect from prolonged air exposure, between 12 and 16 of the 36 clips were duplicated (Table 2). All duplicates were mirrored or at least slightly modified in Adobe Photoshop by increasing their brightness levels to mislead the viewers in assuming all clips were unique. Onto each clip, the true or falsified number of minutes the fish spent on deck exposed to air prior to the reflex test ('air exposure') was labelled, together with a date and time stamp. To falsify air exposure times, an arbitrary 15 min or 30 min were added to the true value. These air exposure periods were chosen to i) represent conventional commercial catch sorting times, and ii) to increase the expectation bias potential for each rater. The greater value (i.e., longer air exposure time) was chosen to increase the likelihood of the rater being influenced by this information if the rater had a preconceived idea about the effect of air exposure on vitality.

**Table 1. List of scoring criteria for categorical reflex responses (i.e., absent, weak, moderate, and strong) of common sole (*Solea solea*) in the order tested within 5 s of observation after stimulus (based on [4, 9]).**

| Reflex | Stimulus | Absent | Weak | Moderate | Strong |
|---|---|---|---|---|---|
| **Body flex** | The fish is held outside the water on the palms of two hands (touching each other) with its belly facing up and its head and tail unsupported. | No active movement, the body rests limp on the hand. | Tail is moving slightly, but not beyond the plain of the hand. | Tail is flexing beyond the plain of the hand. Body may move–spastic flexion; or slowly slipping off the hand. | The fish is actively trying to move head and tail towards each other; or quickly slipping off the hand. |
| **Righting** | The fish is held underwater at the surface on the palms of two hands (touching each other) with its belly facing up and then is slowly released. | Fish drifts and sinks passively to the bottom of the container. | Fish appears stunned, but rights itself very slowly. | Fish appears stunned, but starts to turn after a delay. The rotation can be swift. | Fish actively and quickly turns underwater. |
| **Head** | The fish's head is held between thumb and index finger, with either belly or dorsal side facing up. | No movement. The body dangles motionless. | The fish may move its tail slightly. | The fish may exhibit a cramp-like flexion, but no clear curling, nor repeated bending. | Fish immediately and repeatedly curls around fingers. |
| **Tail grab** | The fish's tail is held between thumb and index finger. | Fish does not struggle free; it remains motionless upon release. | Fish does not struggle free; no swimming movement, but swims away upon release. | Fish does not struggle free, but moves its body as if it attempts to swim away. | The fish actively struggles free and swims away. |

Intensity of a response increases from absent to strong. The speed of a response for weak and moderate categories may be delayed; for strong it should be immediate.

## Data and analyses

To analyse whether a reflex response was biased toward lower or higher tVAS score when an elevated air exposure was falsely indicated on the clip, each rater's scores of duplicate clips were compared with a linear-mixed model (LMMs; lme4 package in R; [28] with as fixed effects: between i) a rater's expectation of an effect of prolonged air exposure on reflex responsiveness, ii) the experience level of each rater, iii) and the reflex type and all possible interactions. Random effects were included for the ID of a given clip, and a rater's ID. The Tukey method was used to compare each reflex for corresponding pairs of duplicate clips shown with either false or true air exposure. These were evaluated by each rater's expectation group (expecting positive, and/or no or negative impact from air exposure) and experience level (no, <100, or ≥100 vertebrate animals previously assessed for reflex responsiveness). A significance level of 0.05 was applied.

Inter- and intra-rater repeatability were estimated based on inter- and intra-rater reliability coefficients [29–30] which were implemented using the irr-package [31]. To estimate inter-rater repeatability, all clips with true air exposure information were included in the dataset (scoring sessions 1–7; Table 2), also arbitrarily stratifying all raters by their reflex rating experience, and calculating across all included clips or specifically per reflex type. To estimate both inter- and intra-rater repeatability on the same dataset, only duplicated clips with true air exposure information were included in the dataset (scoring sessions 5–7; Table 2). The intra-class correlation coefficient (ICC) is based on the ratio of the variability among rater's reflex scores over the sum of this variance plus error, thus ranging between 0 and 100. A higher value of ICC reflects a higher agreement among the raters for a given clip or per reflex type. The ICC measure of association was estimated using the psych package in R [32]. In this study, we report the ICC for a single random rater [29].

## Results

In total, 436 participants scored video clips during the seven dedicated scoring workshops and produced 13,676 scores, because not all participants were equally able to score each of the 36

(a)

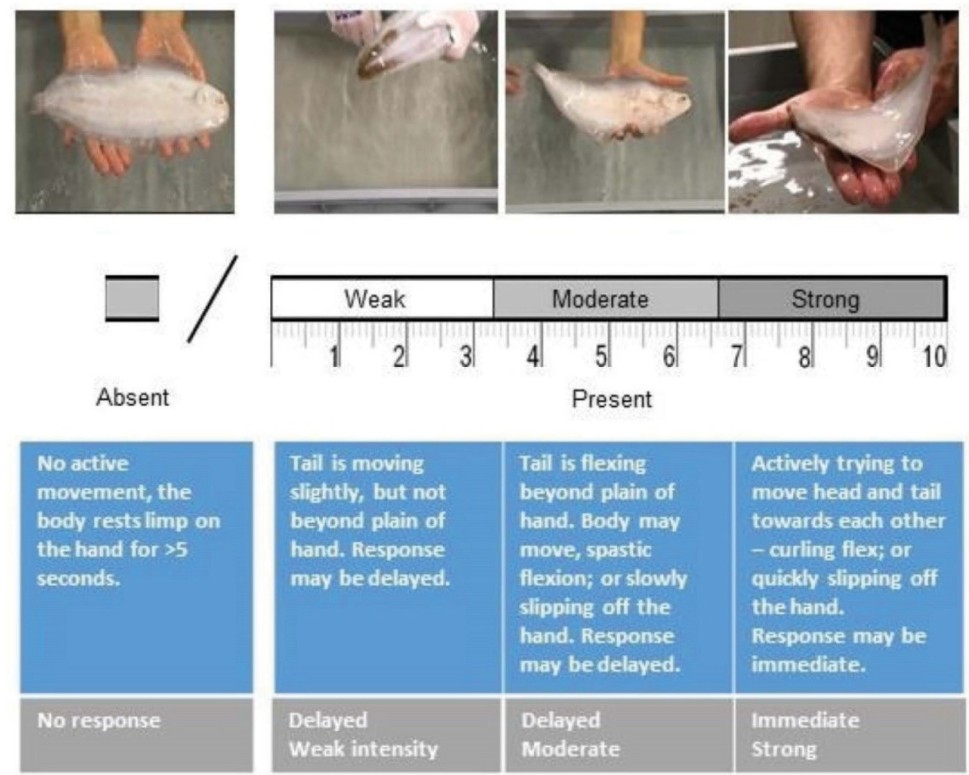

(b)

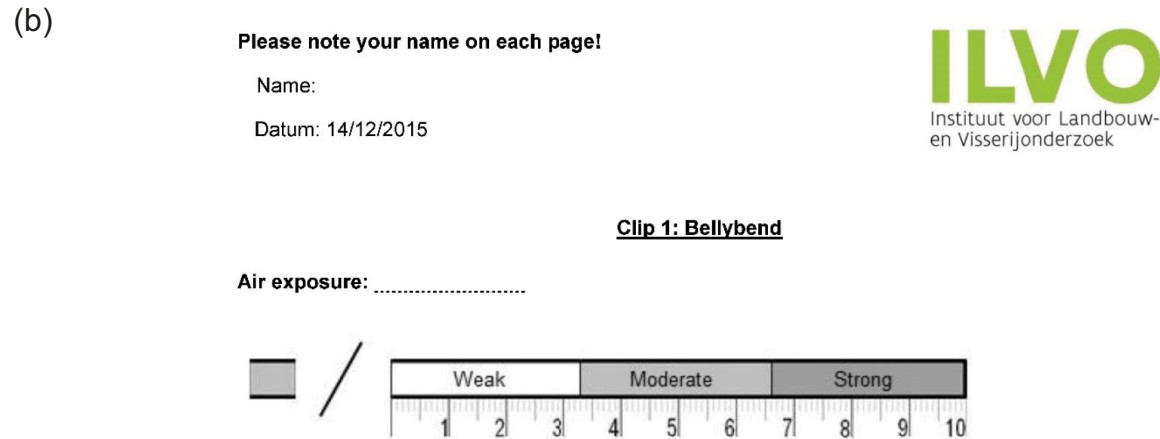

**Fig 2. Example of a pictogram handout (A) and scoresheet (B) to train and score reflexes from video clips.** The scoresheet details how to score the body flex reflex (which was also termed 'bellybend') response of common sole (*Solea solea*) on a continuous tagged-analogue visual scale from a short video clip.

**Table 2. Schematic representation of the treatment (inter-rater vs intra-rater repeatability, with or without expectation bias; see shading) assigned to 36 video clips (each <30 s in length) of a given fish's reflex response per scoring session.**

| Fish | Body flex | Righting | Head | Tail grab |
|---|---|---|---|---|
| | | **Reflex** | | |
| | | **Session 1** | | |
| 1 | F | F | T | F |
| 2 | F | T | F | T |
| 3 | T | T | T | F |
| 4 | T, F | T, F | F, F | T, T |
| 5 | F, F | T, F | F, F | T, T |
| 6 | T, T | F, F | T, F | T, T |
| | | **Sessions 2–4** | | |
| 1 | F | F | T | F |
| 2 | F | T | T | F |
| 3 | T | T | T | F |
| 4 | T, F | T, F | T, F | T, F |
| 5 | T, F | T, F | T, F | T, F |
| 6 | T, F | T, F | T, F | T, F |
| | | **Sessions 5–7** | | |
| 3 | T, F | T, F | T, F | T, F |
| 4 | | T, F | T, F | T, F |
| 5 | | T, T | T, T | T, T |
| 7 | T | T | T | T |
| 8 | T, T | | | |
| 9 | T, T | T, T | T, T | T, T |
| 10 | T, F | | | |

| Inter-Rater Repeatability |
|---|
| Inter-and Intra-Rater Repeatability |
| Inter-and Intra-Rater Repeatability, and Expectation Bias |

The notations 'T,F' or 'T,T' are indicating whether air exposure information was true or falsified on the duplicated video pair, and if not duplicated air exposure was marked as either false or true ('F' or 'T', respectively). For sessions 1–4, if falsified, 15 minutes were added to the true value, for sessions 5–7, 30 minutes were added.

clips (Table 3). The majority (N = 401) were unexperienced raters (i.e., never previously scored animals for reflex responsiveness). Of these, the majority were students, but some were scientists, technicians, observers, and practicing veterinarians/food safety inspectors. Fourteen and 21 participants had scored some (<100 animals) or ≥100 fish (i.e.,'experienced') reflexes before, respectively. One of the raters with some experience had observed behavioural responses among seabirds and seals, but not fish.

## Expectation bias

The dataset that included scores of duplicated clips with either true or falsified air exposure information comprised 3,525 scores which were assigned in workshop sessions 2–7 to duplicate clips by those participants who indicated an expectation about the effect of air exposure on reflex responsiveness (Table 2). Scores by participants from session 1 were not included, because not all duplicated clips were paired by true/false air exposure information (Table 2). Based on histogram data indicating a clear distinction at greater and less than 30, a positive

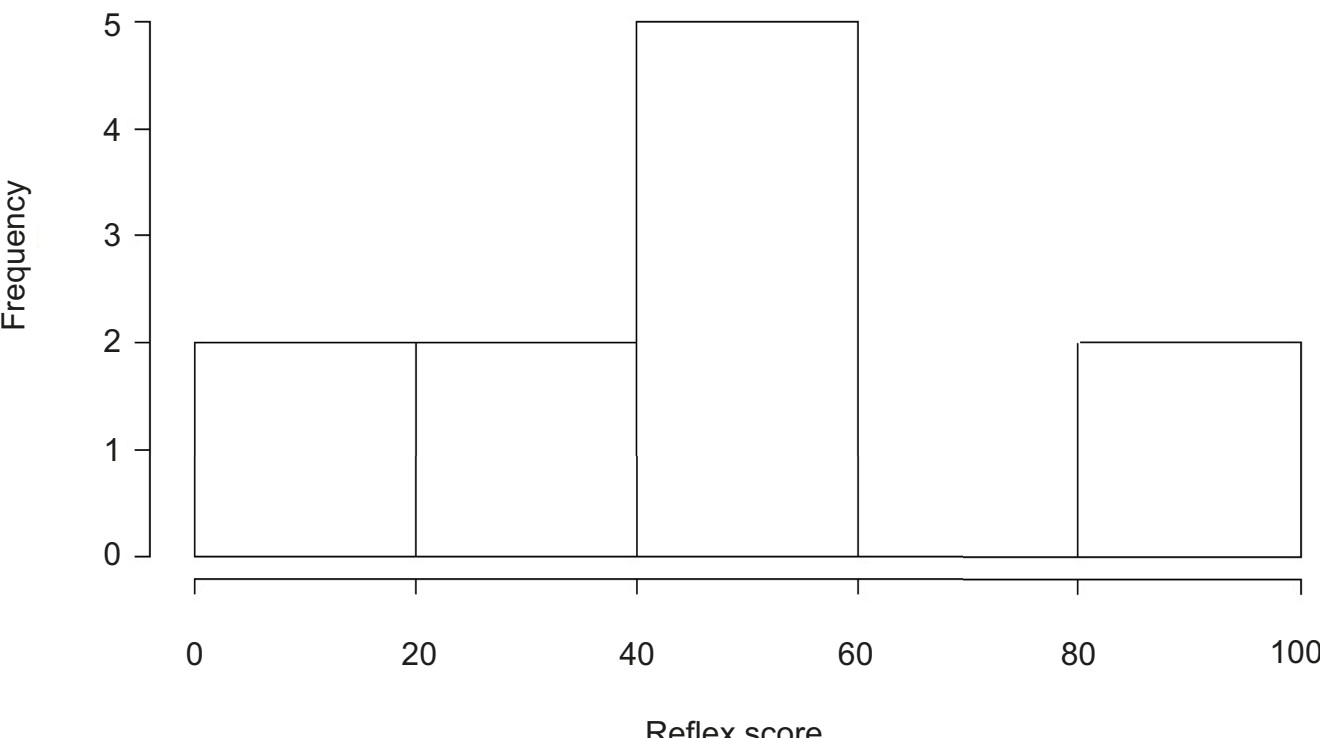

**Fig 3. Frequency distribution of average 'silver standard' reflex scores.** Scores by three expert raters who developed the reflex scoring methodology and were experienced raters were considered as the 'silver standard'. These three raters scored all original clips (N = 12) that went into the making of the scoring video for sessions 2–4.

expectation (i.e., air exposure would exacerbate reflex impairment) was set at <30, and a negative expectation (i.e., air exposure would reduce reflex impairment) was ≥30 (Fig 4). Of these scores, 70% were scored with a positive expectation by the participant (Fig 4). An expectation of the effect of air exposure on reflex responsiveness did not bias the scoring of reflex clips. The null hypothesis (i.e., no difference in scores due to air exposure information) was not rejected for raters who expected air exposure to positively affect reflex impairment (N = 128; Table 3). Overall, these raters were not more likely to assign a lower score to a duplicated clip that showed falsified air exposure (extra 15 or 30 min) compared to the original, which was stamped with the true air exposure time (Table 4). Generally, where available, the median scores followed what the three expert raters had assigned to each clip ('silver standard' score), although for some clips scored by raters with some or experienced raters, their median values were off the mark compared to the silver standard (Fig 5).

Nevertheless, for some duplicated clips that were scored by raters with some prior reflex scoring experience, lower scores were assigned to clips as postulated by our null hypothesis (i.e., duplicates with IDs 3_10a & b; Fig 5C). But this difference was not significant (Table 4). In advance of scoring, some raters expected that the reflex would not be affected by air exposure or would even become stronger (N = 85; decreased impairment = negative expectation, Table 3). This aligned with clips of the body flex reflex, for which raters with no reflex assessment experience consistently scored higher for clips with falsified compared with true air exposure (Table 4; Fig 5B). This contrasted our null hypothesis.

**Table 3. Number, gender, experience, and expectation of workshop participants per scoring session (1–7), stratified by previous experience in scoring reflex responsiveness of live animals ('none': no animals scored; ' some': <100 animals scored; and 'experienced': ≥100 animals scored).**

| Session | No. participants | Male | Female | NA | Experience | Expectation | | |
|---|---|---|---|---|---|---|---|---|
| | | | | | | Positive | Negative | NA |
| **1** | 157 | 35 | 120 | | None | 54 | 62 | 39 |
| | | 2 | 0 | | Experienced | 2 | 0 | 0 |
| **2** | 18 | 8 | 7 | | None | 3 | 12 | 0 |
| | | 3 | 0 | | Experienced | 1 | 2 | 0 |
| **3** | 13 | 2 | 1 | | None | 2 | 1 | 0 |
| | | 3 | 1 | | Some | 3 | 1 | 0 |
| | | 3 | 3 | | Experienced | 1 | 5 | 0 |
| **4** | 19 | 4 | 0 | | None | 4 | 0 | 0 |
| | | 5 | 3 | | Some | 7 | 1 | 0 |
| | | 4 | 3 | | Experienced | 5 | 2 | 0 |
| **5** | 182 | 39 | 140 | 2 | None | 78 | 41 | 62 |
| | | 1 | 0 | | Experienced | 1 | 0 | 0 |
| **6** | 33 | 20 | 12 | | None | 13 | 16 | 3 |
| | | 1 | 0 | | Experienced | 1 | 0 | 0 |
| **7** | 14 | 4 | 6 | 1 | None | 7 | 3 | 1 |
| | | 2 | 0 | | Some | 1 | 1 | 0 |
| | | 1 | 0 | | Experienced | 1 | 0 | 0 |
| **Total** | 436 | 134 | 296 | 3 | | 184 | 147 | 105 |

'Expectation' was classified based on the rater's response to a question on the scoresheet asking whether s/he expected air exposure to impact reflex impairment, either positively or negatively (i.e., the rater believes that prolonged air exposure would exacerbate or reduce reflex impairment, respectively). NA, not all participants revealed their gender or gave a score for the expectation question.

## Intra- and inter- rater repeatability

When quantifying inter-rater repeatability (dataset included scores of all clips with true air exposure information, some clips were duplicated; 6,664 observations), raters with different experience levels in scoring reflex impairment were able to reproduce the same score for a given clip when scored independently in different scoring sessions with an intra-class correlation coefficients of 76% (68% 84%, lower and upper confidence interval, CI). Participants who had no prior scoring experience produced a lower intra-class correlation coefficient (ICC = 76%, 68% 84% CI) compared with participants who had scored at least some fish throughout their career (ICC = 79%, 71% 87% CI). However, the latter sample size was rather small (N = 29) compared to 396 raters with no experience who were considered in this analysis.

A similar pattern resulted when comparing ICC values per reflex type. For example, for the tail grab reflex, raters with at least some experience scored more consistently than raters with no experience (ICC = 86%, 76% 94% upper and lower CI vs ICC = 81%, 69% 92% upper and lower CI, respectively). Similarly, but with a less prominent difference, for the head reflex, raters with at least some experience had an ICC value of 79% (63% 93% lower and upper CI) compared to 78% (61% 93% CI) for raters with no experience. Including only seagoing observers and those experts who developed this methodology, increased the ICC (ICC = 83%, 67% 95% upper and lower CI).

However, in contrast, for the righting reflex, the pattern was reversed: raters with no experience scored more consistently than raters with experience (ICC = 71%, 50% 92% upper and lower CI versus ICC = 49%, 27% 83% upper and lower CI, respectively). The least repeatable

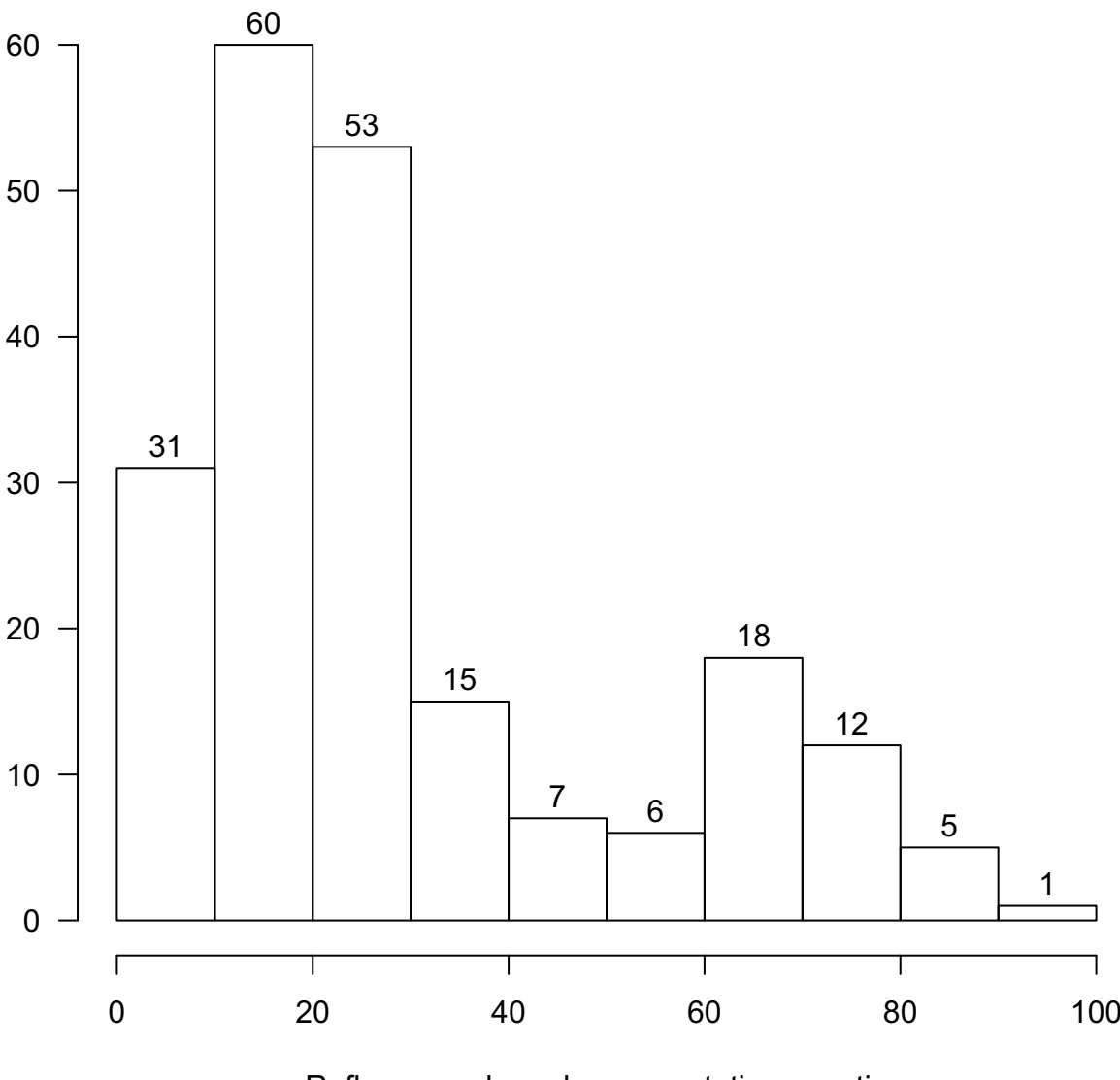

**Fig 4. Frequency distribution of hypothetical tVAS scores provided by unique raters from scoring sessions 2–7.** N of unique raters is indicated above each bar as it was marked by a rater on their scoresheet in response to a question whether a reflex response would weaken or strengthen when the animal was knowingly exposed to air for a prolonged period (15–30 min).

were the scores for the body flex, regardless of experience (ICC = 15%, 0.07% 46% upper and lower CI versus ICC = <1%, -11% 37% upper and lower CI, respectively for raters with none or at least some experience).

The dataset which included duplicated clips with only true air exposure information, to calculate ICC of both intra- and inter-rater reliability comprised 3,664 observations. Across all reflexes, relatively high ICC values of 74% were achieved for inter- and intra-rater reliability, for both. For individual reflexes, highest ICC of both intra- and inter-rater reliability (for both the values were almost the same and differed from beyond the third decimal) were achieved for head (92%), tail grab (78%), righting (45%), and by far the lowest ICC was achieved for body flex (<1%).

**Table 4. Tukey comparisons of the least-square mean (lsmean) ± SE reflex score of a given reflex type which was scored by a rater with a certain experience and a positive (1) or negative (0) expectation.**

| Experience | Reflex | Expectation | Air exposure | lsmean | SE | l.CL | u.CL | Group |
|---|---|---|---|---|---|---|---|---|
| **None** | Body flex | 1 | TRUE | 50.8 | 15.4 | 20.6 | 81 | a |
| | | 1 | FALSE | 58.7 | 15.4 | 28.5 | 88.9 | b |
| | | 0 | TRUE | 55.2 | 15.4 | 25.0 | 85.5 | ab |
| | | 0 | FALSE | 60.4 | 15.4 | 30.2 | 90.7 | b |
| | Head | 1 | TRUE | 39.8 | 13.4 | 13.6 | 66 | a |
| | | 1 | FALSE | 37.0 | 13.4 | 10.8 | 63.2 | a |
| | | 0 | TRUE | 40.8 | 13.4 | 14.6 | 67.1 | a |
| | | 0 | FALSE | 38.2 | 13.4 | 12.0 | 64.4 | a |
| | Righting | 1 | TRUE | 46.3 | 13.4 | 20.1 | 72.5 | a |
| | | 1 | FALSE | 49.4 | 13.4 | 23.2 | 75.7 | a |
| | | 0 | TRUE | 48.3 | 13.4 | 22.1 | 74.5 | a |
| | | 0 | FALSE | 51.3 | 13.4 | 25.0 | 77.5 | a |
| | Tail grab | 1 | TRUE | 46.0 | 13.4 | 19.8 | 72.2 | a |
| | | 1 | FALSE | 45.1 | 13.4 | 19.0 | 71.3 | a |
| | | 0 | TRUE | 47.6 | 13.4 | 21.4 | 73.8 | a |
| | | 0 | FALSE | 49.0 | 13.4 | 22.8 | 75.2 | a |
| **Some** | Body flex | 1 | TRUE | 60.8 | 16.2 | 29.0 | 92.5 | ab |
| | | 1 | FALSE | 69.2 | 16.2 | 37.4 | 100.9 | b |
| | | 0 | TRUE | 42.9 | 17.7 | 8.2 | 77.6 | a |
| | | 0 | FALSE | 53.9 | 17.7 | 19.2 | 88.6 | ab |
| | Head | 1 | TRUE | 42.3 | 13.7 | 15.4 | 69.1 | a |
| | | 1 | FALSE | 34.1 | 13.7 | 7.3 | 61 | a |
| | | 0 | TRUE | 39.4 | 14.8 | 10.3 | 68.4 | a |
| | | 0 | FALSE | 31.9 | 14.8 | 2.8 | 60.9 | a |
| | Righting | 1 | TRUE | 62.5 | 13.7 | 35.7 | 89.4 | a |
| | | 1 | FALSE | 51.8 | 13.9 | 24.6 | 79.1 | a |
| | | 0 | TRUE | 61.9 | 14.8 | 32.9 | 91 | a |
| | | 0 | FALSE | 50.2 | 15.2 | 20.3 | 80 | a |
| | Tail grab | 1 | TRUE | 53.0 | 13.7 | 26.2 | 79.9 | a |
| | | 1 | FALSE | 54.5 | 13.7 | 27.6 | 81.3 | a |
| | | 0 | TRUE | 51.0 | 14.8 | 21.9 | 80 | a |
| | | 0 | FALSE | 52.1 | 14.8 | 23.1 | 81.2 | a |
| **Experienced** | Body flex | 1 | TRUE | 66.1 | 16.2 | 34.4 | 97.9 | a |
| | | 1 | FALSE | 62.6 | 16.2 | 30.8 | 94.4 | a |
| | | 0 | TRUE | 74.0 | 16.5 | 41.7 | 106.3 | a |
| | | 0 | FALSE | 75.8 | 16.5 | 43.5 | 108.2 | a |
| | Head | 1 | TRUE | 41.1 | 13.8 | 13.9 | 68.2 | a |
| | | 1 | FALSE | 39.4 | 13.8 | 12.3 | 66.5 | a |
| | | 0 | TRUE | 40.7 | 13.8 | 13.7 | 67.7 | a |
| | | 0 | FALSE | 38.7 | 13.8 | 11.7 | 65.7 | a |
| | Righting | 1 | TRUE | 53.1 | 13.9 | 26.0 | 80.3 | a |
| | | 1 | FALSE | 51.2 | 14.0 | 23.8 | 78.7 | a |
| | | 0 | TRUE | 57.0 | 13.8 | 30.0 | 84 | a |
| | | 0 | FALSE | 50.9 | 14.0 | 23.5 | 78.3 | a |
| | Tail grab | 1 | TRUE | 57.5 | 13.8 | 30.3 | 84.6 | a |
| | | 1 | FALSE | 51.6 | 13.8 | 24.5 | 78.8 | a |

*(Continued)*

**Table 4.** (Continued)

| Experience | Reflex | Expectation | Air exposure | lsmean | SE | l.CL | u.CL | Group |
|---|---|---|---|---|---|---|---|---|
| | | 0 | TRUE | 56.4 | 13.8 | 29.4 | 83.4 | a |
| | | 0 | FALSE | 48.5 | 13.8 | 21.5 | 75.5 | a |

Clips were duplicated within a scoring video and imprinted onto the screened clip with either false (an added 15 or 30 min to the true value) or true air exposure information. A rater's expectation (scored on a scale of 0 to 100) of the effect of prolonged, onboard air exposure on a fishes' reflex responsiveness was categorized as to whether it would result in either a weaker (<30; positive expectation; 1) reflex response or no effect (≥30, no or negative/wrong expectation; 0). Our hypothesis was that clips imprinted with false air exposure information would receive a lower score than their duplicate shown with the true value, as the fish would have been weakened from additional air exposure (positive expectation). Groups with the same letter were not significantly different at $p$ = 0.05.

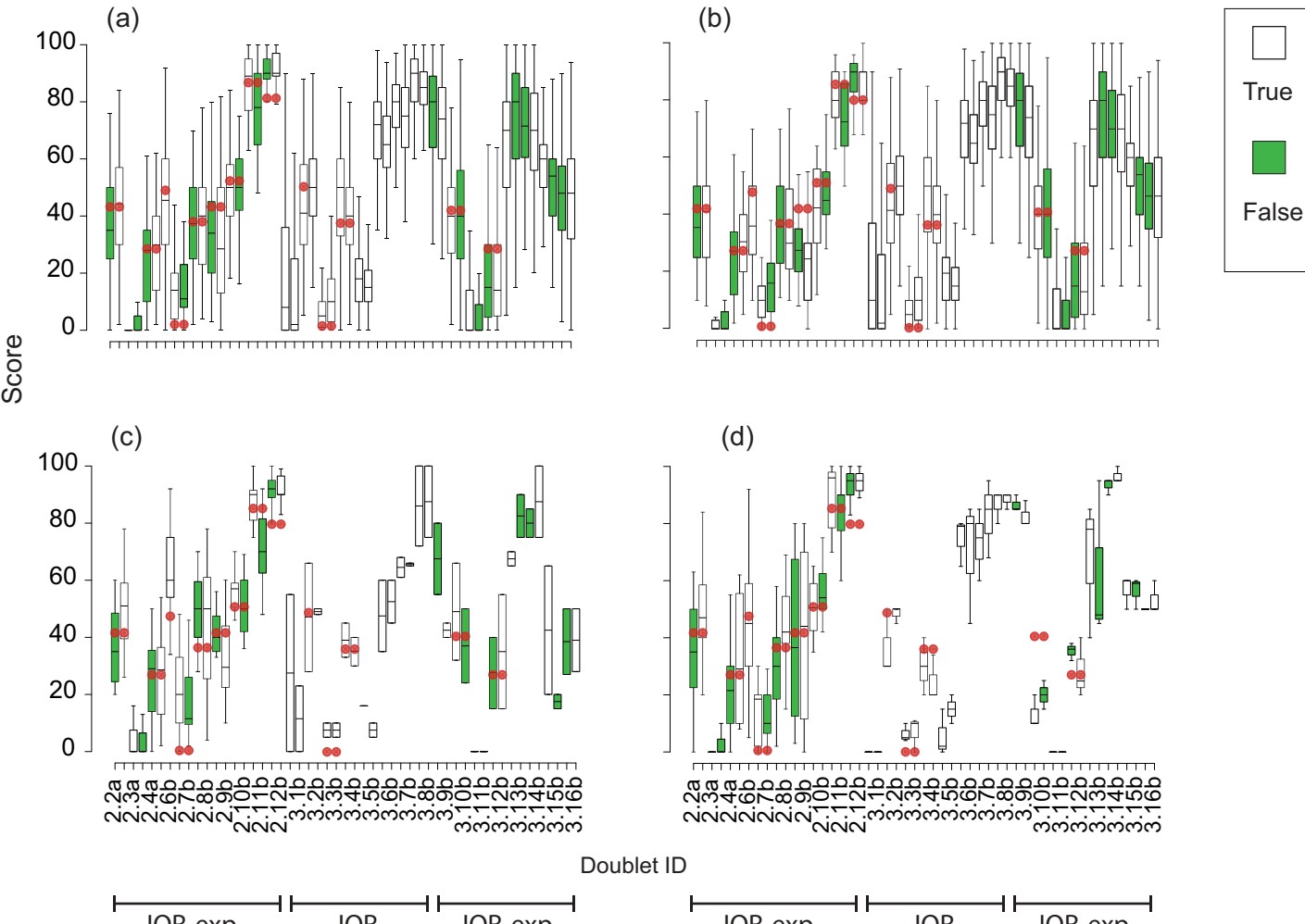

**Fig 5. Plot of mean score per video clip doublet (with either 'true' or 'false' air exposure information) of a given reflex across all workshop participants (a), and then stratified by experience in scoring reflexes of fish: none (b); some (c); and experienced (d).** Doublet ID includes a scoring video clip ID (2 or 3) and a running ID number for each doublet, with each clip of a doublet abbreviated by 'a' or 'b'. Treatments include: 'Intra-rater reliability with expectation bias' (IOR-exp.), which refers to duplicated clips of the same fish and reflex with either true or false air exposure information; or 'Intra-rater reliability' (IOR), which refers to duplicated clips of the same fish and reflex and always true air exposure information. Where available, dots indicate the 'silver standard' scores which were averaged across three experienced, expert raters who scored 12 unmodified, original clips.

## Discussion

There is a global effort to determine the limitations and strengths of methods that profile fish condition related to fishing impacts and survival prediction [21, 33]. This study examined whether vitality information is reliable based on the involvement of multiple raters and/or on their experience level, and whether scoring repeatability can be influenced by knowing the treatment a fish has received. Results suggest that vitality assessments using reflex responsiveness are robust.

In regard to expectation bias, there was no evidence that the exaggerated air exposure information influenced intra-rater repeatability. Regardless of their experience, raters were not misled to assign lower reflex scores to fish which they believed were exposed to air for a prolonged period of time, even when they expected air exposure to positively impact reflex impairment. This does not mean that other variables cannot invite expectation bias; however, it does suggest that perhaps when focusing on a specific metric over a short time frame, the rater does not subconsciously bias their assessment, especially when the scoring criteria (here between absent and present) are unambiguous.

These results are promising if reflexes are to be used in settings with multiple, independent raters and/or with raters who do not have a strong background in reflex assessment. We do however acknowledge that this study was done through video clip analysis rather than having participants handle fish. There is the possibility that tactile experience in fish handling or reflex scoring could result in inter-rater variability among scores. However, [9] found no inter-rater differences when multiple participants scored the same live fish for reflex impairment.

While rater experience in conducting reflex assessments did not bias the scoring outcomes (similar to results from [9]), results suggest that bias is potentially more likely to be introduced through subjective reflexes than raters; especially when reflexes were to be presented as <30 sec long video clips. This includes reflexes that are difficult to assess or that elicit responses that are difficult to discern between presence and absence; or reflexes such as body flex and righting which during evaluation were rapidly tested in succession of each other. This supports the need for researchers to scrutinize the selected reflexes that will be used for a vitality study in advance of data collection based on a screening for consistent and unambiguous candidate reflexes among unstressed fish [4], to be deliberate about scoring metrics (i.e., binary vs. continuous scoring), and ideally, establish a concrete physiological link between a stressor and reflex impairment to validate underlying hypotheses that such links exist [20–21]. Ideally, during data collection, each rater should be blinded and unaware about any prior treatments a study animal may have received, likewise an analyst should be unaware of who did the scoring [15]. Attention has to be paid when editing video clips accordingly. Experience may contribute to a subjective interpretation of scoring criteria, when pre-gained routines and self-made 'rules' may bias an assessment.

This study also supports the use of video-taped reflex assessments that can be reviewed at a later time. This has implications for allowing multiple assessments of the same video and to include reviewers who are unable to go to sea for each field trial. It also is beneficial for training purposes to minimize handling of fish. While there is evidence that untrained raters are capable of rating as or even more accurately as experienced raters, for future studies using reflex impairment as a vitality metric, we recommend having a substantial training programme for raters which includes protocols with clear and meaningful definitions, scoring of videos with pictogram-based handouts, repetitive training sessions and continued repeatability checks [34]. In addition, if video assessments are performed, it is helpful to have sheets describing the reflexes in front of the raters, constraining a fixed amount of time to observe each clip, and ensuring only one reflex is shown in a video clip at a time. There is also the potential to have a

video shown on a touch screen where the rater could be more in control of viewing; however, time to review should be limited.

Blinding and intra- and inter-rater reliability analyses are relevant concepts which should be considered for robust inference within experimental fisheries science, especially where many independent raters are involved. For example, when fish otolith are read for their age (e.g., [35]) or when using vitality indices to evaluate welfare and/or freshness of catches either on-board vessels or at fish auctions. Among domestic farm animals, such assessments are routinely done (e.g., [36]).

## Acknowledgments

We thank Frank Tuyttens and Eva Van Laer for their conceptual input and feedback on this project, and Ruben Theunynck, Leonie Jacobs, Lisanne Stadig, and Lancelot Blondeel for organizing some of the workshops. We are grateful to Maarten Soetaert for overseeing the ethically correct handling of beam-trawled flatfish for this study. We thank the expert working group on Methods for Estimating Discard Survival (WKMEDS, now WGMEDS) of the International Council for the Exploration of the Sea (ICES) for facilitating this research, and Tom Catchpole for inviting colleagues from Cefas to take part in one of the workshops. The contributions by all workshop participants are greatly acknowledged. We thank Christian Vanden Berghe, Eddy Buyvoets, staff from the Flanders Marine Institute, the crew of a Belgian beam trawler and R/V *Simon Stevin* for helping to catch some of the flatfish which were video-taped for this research. We are grateful to Shannon Fitzgerald, Jordan Watson, and the anonymous reviewers for their constructive comments on an earlier version of this manuscript. The scientific results and conclusions, as well as any views or opinions expressed herein, are those of the authors and do not necessarily reflect those of NOAA or the Department of Commerce.

## Author Contributions

**Conceptualization:** Sven Sebastian Uhlmann, Noëlle Yochum.

**Data curation:** Sven Sebastian Uhlmann, Bart Ampe.

**Formal analysis:** Sven Sebastian Uhlmann, Bart Ampe.

**Funding acquisition:** Sven Sebastian Uhlmann.

**Investigation:** Sven Sebastian Uhlmann, Noëlle Yochum.

**Methodology:** Sven Sebastian Uhlmann, Noëlle Yochum.

**Project administration:** Sven Sebastian Uhlmann.

**Resources:** Sven Sebastian Uhlmann.

**Software:** Bart Ampe.

**Validation:** Sven Sebastian Uhlmann, Noëlle Yochum, Bart Ampe.

**Visualization:** Sven Sebastian Uhlmann, Bart Ampe.

**Writing – original draft:** Sven Sebastian Uhlmann, Noëlle Yochum.

**Writing – review & editing:** Sven Sebastian Uhlmann, Noëlle Yochum, Bart Ampe.

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
