## [Decision Letter · Decision Letter 0]

21 Jan 2020

PONE-D-19-34394

Repeatability of flatfish reflex impairment assessments based on video recordings

PLOS ONE

Dear Dr Uhlmann,

Thank you for submitting your manuscript to PLOS ONE. After careful consideration, we feel that it has merit but does not fully meet PLOS ONE’s publication criteria as it currently stands. Therefore, we invite you to submit a revised version of the manuscript that addresses the points raised during the review process.

Editors comments: The reviewer, while enthusiastic, outlines a number of ways in which the manuscript could be improved. Please pay attention to these in particular, the following three points

The intro needs a paragraph to contextualize what exactly is a reflex impairment score and how it relates to physiological status in the fish

The discussion needs some work, particularly around comparisons to other work.

I also suggest that a table outlining general recommendations for avoiding bias’s in an experimental design so that others can avoid the same pitfalls and a good discussion around this.

We would appreciate receiving your revised manuscript by Mar 06 2020 11:59PM. To enhance the reproducibility of your results, we recommend that if applicable you deposit your laboratory protocols in protocols.io, where a protocol can be assigned its own identifier (DOI) such that it can be cited independently in the future. For instructions see: http://journals.plos.org/plosone/s/submission-guidelines#loc-laboratory-protocols

We look forward to receiving your revised manuscript.

Kind regards,

Judi Hewitt

Academic Editor

PLOS ONE

Journal Requirements:

2. We note that Figure 1 includes an image of a [patient / participant / in the study]. 

Reviewers' comments:

Reviewer's Responses to Questions

**Comments to the Author**

1. Is the manuscript technically sound, and do the data support the conclusions?

Reviewer #1: Yes

2. Has the statistical analysis been performed appropriately and rigorously? 

Reviewer #1: Yes

3. Have the authors made all data underlying the findings in their manuscript fully available?

Reviewer #1: Yes

4. Is the manuscript presented in an intelligible fashion and written in standard English?

Reviewer #1: Yes

5. Review Comments to the Author

Reviewer #1: The work presented by Uhlmann et al. was quite interesting and highly relevant given the increased usage of reflex scores in assessing fish stress status in numerous contexts and settings. Overall, it was quite an enjoyable read and thought the concept and data were very interesting. Although, I have some minor concerns that I would like to see addressed before acceptance of the manuscript.

• Weird spacing in the intro. Eg lines 81-83, 62-63

• The intro overall conveys a very good and concise of the literature regarding biases and issues with using reflex impairment scores. However, I think some additional information is need to contextualize what exactly is a reflex impairment score and how it relates to physiological status in the fish. Readers unfamiliar with the topic might be confused as to why this metric is even used. I recommend just including a short paragraph outlining this.

• Backing up my point, in the objectives, the authors are indicating using real vs fake air exposure info. If the reader did not understanding the underlying physiological processes associated with air exposure (i.e. hypoxia, metabolic acidosis, substrate depletion, etc.) then making the connection to a reflex score might be difficult.

• The video/crowd sourcing data idea is really cool! I commend the authors on thinking outside the box on this one.

• Table 7: may wish to include a reference indicating where this information was originally sourced from? In the methods it does indicate that it uses standardized metrics employed by fisheries science with this species.

• Line 158, ok but how do the authors know that these reflexes are characterized as such? Was one person quantifying across all of the videos to indicate that this was a representative “strong” or “weak” fish. I would just like some more details pertaining how the representative videos were selected in the first place especially if observer bias is a problem.

• Line 167, why were these durations used? It seems unlikely to have a fish air exposed in excess of 30 min.

• Table 2: why was gilling, a standard part of the ramp score, no used in the reflex assessments?

• Were the veterinarians in the study strictly aquatic vets or was it a mix of students studying general vertebrate anatomy/physiology. I ask as this as this in itself may pose an issue relating to experience working with fish vs non-fish vertebrates.

• I realize there is a large volume of data but would it be worth visualizing some of it in the form of a graph. May help the readers get a better sense of the trends in the data as the results section is pretty text heavy which I totally understand is hard to otherwise with a data set like this one. I just think a couple visuals would go a long way in conveying the data.

• Overall, I found the discussion quite short and fairly superficial. I recommend incorporating more comparisons to other work especially in the realm of observer bias in animal behaviour for which there are numerous works on. The authors have a lot of data here with some pretty neat trends but don’t really discuss their data to any great length.

• It may be worth having either a table or a paragraph outlining general recommendations for avoiding bias’s in an experimental design so that others can avoid the same pitfalls. This is really what this paper is about so I recommend discussing this in some great detail here.

• How would these results compare to say taking reflex scores in a field setting (i.e. without camera) where the observer may only have a single opportunity to view a fish’s behaviour?

• Outside of sole, how applicable would these results be other species and settings? Presumably, the various body characteristics of the fish may also help enhance/impair scoring especially if the behaviour is relatively subtle. You may wish to highlight context/species specific effects in your discussion as something like a salmon or a shark may not have the same observer bias associated with it.

6. PLOS authors have the option to publish the peer review history of their article (what does this mean?). If published, this will include your full peer review and any attached files.

Reviewer #1: No

---

## [Author Response · Author response to Decision Letter 0]

28 Jan 2020

Journal requirement: naming of files

As requested we have paid attention to the file and formatting style and naming requirements as outlined in the provided guides for authors documents. Accordingly, adjustments to the title page were made. Please note that the formatting of the affiliations are inconsistent in the example provided (i.e., there seem to be a space missing between superscript and affiliation address in some instances, check 1 Department, Institution, City, State, Country vs 2).

Journal requirement: figure 1 – license to publish

Although all persons pictured were facing with their backs to the camera and the purpose of this image is to illustrate the projection of the video clips onto a big screen and not to show the audience or individual participants, we have blurred the audience in photoshop to make sure that no one is identifiable and added this as a comment to its figure caption (L123-L124).

Reviewer #1: 

Comment 1 – Weird spacing in the intro. We have corrected the weird spacing and apologise for it. It was an artefact from sharing a manuscript draft between North American and European Microsoft Office Word packages, which caused compatibility issues with the formatting, and eventually I created a new version by copying and pasting.

2 – Additional information needed in intro. As requested, we included a short paragraph at the beginning of the introduction outlining the concept of reflex scoring and why it is used (L47-L55). Note that the numbering of in-text citations has changed to accommodate for extra referencing of studies.

3 – Link between hypoxia and impairment. See comment 2 above. This paragraph also mentions how the exposure to air from hypoxia could contribute to impaired reflex responsiveness (L53-L55).

4 – Video/crowd sourcing idea. We thank the reviewer for this appreciation of our efforts. It all started off as a ‘pet project’, and soon we realized that we could make a very useful contribution to the study of reflex responsiveness of fish and experimental design in fisheries science in general.

5 – Table 7. We are not sure whether the reviewer refers to Table 1 here, because there is no Table 7? We have added another reference to the caption of Table 1 (L153). 

6 – Selection of clips. We have added a sentence to explain how clips were selected and added a histogram of the distribution of averaged ‘silver standard’ scores (Fig 3). These were based on scores by three experts, who were experienced raters who were involved in developing the methodology and who scored 12 unmodified, original clips in a separate video. These 12 clips were then used to create the scoring video by introducing duplicates, and falsifying air exposure information. We also modified Figure 4 (now Figure 5), to indicate how the percentile ranges of pooled raters (by experience) compared to the ‘silver standard’ scores averaged across the three expert raters (L305-L307). 

7 – Choice of fictive air exposure periods. These air exposure periods were chosen for two reasons: to represent commercial fishing practices (depending on catch volumes which can be large), it may take at least 15 min to 30 min until all air-exposed catch has been sorted by the crew. Secondly, the longer air exposure period was chosen to increase any expectation potential (L187-L189).

8 – Choice of reflexes. If the reviewer refers to what is called “head complex” or “operculum” in the literature and scores the functioning of the breathing apparatus, then this reflex was not selected, because it was considered that it may be difficult to discern for a rater whether the gills/operculum and/or mouth of a sole would be moving from a video clip.

9 – Veterinarians in the study. In the material and Methods section, we have specified that experience can refer to having assessed any vertebrate animal for reflex behaviours (L212-L213), and mentioned in the Results section how many of the raters with some experience have had experience with scoring behavioural responses among vertebrate animals other than fish (L235-L236). None of the practicing veterinarians had indicated any prior experience with scoring reflexes.

10 – Visualize some of the data. We suggest to include as an online supplement a video which we produced that includes a powerpoint lecture, training video and commentary. Other than that, we think that the structure and experimental design was sufficiently and succinctly summarized in the tables we provided. But we also included another histogram to illustrate the distribution of ‘silver standard’ scores (see comment 6).

11 – Superficial discussion. In places, we have revised the discussion to relate our results in greater depth to recent and relevant literature (L381-L383; L404-L406). 

12 – Additional table or paragraph outlining general recommendations. We think that our concluding and several other paragraphs in the discussion list our recommendations, and we do not see the necessity to add another table to keep this article more concise.

---

## [Editor Report · Decision Letter 1]

7 Feb 2020

Repeatability of flatfish reflex impairment assessments based on video recordings

PONE-D-19-34394R1

Dear Dr. Uhlmann,

We are pleased to inform you that your manuscript has been judged scientifically suitable for publication and will be formally accepted for publication once it complies with all outstanding technical requirements.

With kind regards,

Judi Hewitt

Academic Editor

PLOS ONE
---

## [Editor Report · Acceptance letter]

18 Feb 2020

PONE-D-19-34394R1 

Repeatability of flatfish reflex impairment assessments based on video recordings 

Dear Dr. Uhlmann:

I am pleased to inform you that your manuscript has been deemed suitable for publication in PLOS ONE. Congratulations! Your manuscript is now with our production department. 

With kind regards,

on behalf of

Dr. Judi Hewitt 

Academic Editor

PLOS ONE